# The Effect of 24 h Shift Work on the Nutritional Status of Healthcare Workers: An Observational Follow-Up Study from Türkiye

**DOI:** 10.3390/nu16132088

**Published:** 2024-06-29

**Authors:** Semra Navruz Varlı, Hande Mortaş

**Affiliations:** Department of Nutrition and Dietetics, Faculty of Health Sciences, Gazi University, Ankara 06490, Türkiye; handeyilmaz@gazi.edu.tr

**Keywords:** 24-h shift, night shift, long working hours, health workers, energy, macronutrient intake, micronutrient intake, vitamins, minerals

## Abstract

This study aimed to determine changes in energy and nutrient intakes over three consecutive days, including the day of the shift, and the days before and after the shift, in healthcare workers working in a 24 h shift system. This study is an observational follow-up study conducted with a total of 500 volunteer healthcare professionals. Food consumption records were taken over 3 consecutive days: pre-shift (off day), shift day (24 h shift), and post-shift (off day). Mean daily intakes of energy, carbohydrate, protein, fat, saturated fat, caffeine, vitamins B_1_, B_2_, niacin, B_6_, folate, and B_12_, potassium, magnesium, phosphorus, iron, and zinc are listed from highest to lowest as shift day > pre-shift > post-shift (*p* < 0.05 for all pairwise comparisons). While fiber, vitamin C, and calcium intakes were similar on the shift day and pre-shift day, they were significantly lower on the post-shift day (*p* < 0.05). The lowest dietary reference intake percentages on the post-shift day were calcium, fiber, and folate, respectively. In the present study, significant differences were detected in the energy, micronutrient-intake, and macronutrient-intake levels between the pre-shift day, shift day, and post-shift day of healthcare workers. Awareness should be increased regarding the decreased nutrient intake seen especially on the first day after a 24 h shift, and appropriate precautions should be taken to increase calcium, fiber, and folate intake levels.

## 1. Introduction

Shift work is a working system created to provide uninterrupted service at any time of the day, including working periods outside of daytime working hours (7:00/8:00 a.m.–5:00/6:00 p.m.) [1,2]. The shift work system is applied for continuity of service, especially in hospitals and other health institutions, as well as in many basic areas such as security and transportation. Although this working system is indispensable for ensuring continuity in the provision of health services to society, working outside routine working hours during the day, especially the night shift, negatively affects the health of employees in physiological, psychological, and social terms [1,3].

In the most basic terms, individuals working in shifts are forced to perform various vital activities, such as working and eating, in the dark part of the day, which they would do in daylight, according to their biological rhythm. In other words, since working the night shift brings about sleeping during the daytime, it disrupts the biological rhythm of the healthcare worker, along with the change in nutrition and sleep patterns. This biological rhythm, called the circadian rhythm, is regulated by exogenous factors, such as light, nutrition, social behavior, work life, and some endogenous factors, in healthy individuals [4]. Disruption of the circadian rhythm leads to the development of many diseases such as obesity, metabolic syndrome, and cancer [5]. In addition to the disruption of the circadian rhythm, healthcare workers working night shifts frequently experience gastrointestinal system disorders such as constipation, digestive problems, and various mood disorders [3,6,7].

Mood disorders, such as depression, anxiety, stress, and burnout, have been shown to affect food preferences. It is thought that dietary preferences change in shift workers due to the combined effect of mood disorders and decreased sleep time [8]. Short rest periods during shift-working hours and inadequate and limited food choices may also cause changes in nutritional intake [9,10,11]. In addition to these factors, nutritional intake and quality in healthcare personnel working in shifts may be affected by gender, body mass index (BMI), shift work history, work performed, and demographic factors [8]. The quality and safety of health services are highly related to the health, efficiency, and capacity of health professionals [12]. For this reason, it is important to investigate the effects of the night-shift working system, which directly concerns healthcare workers’ health, on employees’ nutritional status. Thus, comprehensive corrective actions can be planned to take the necessary precautions and reduce health risks.

When studies in the literature investigating the effect of the night-shift working system on nutrition in healthcare workers are examined, it can be seen that the differences in the energy and nutrient intakes of day workers versus shift workers, or shift days versus off days, are often evaluated [9,10,13,14]. To the best of our knowledge and as far as we have researched, no study has been found that evaluates, in detail, nutritional intake on the day immediately following tiring and long (24 h) shifts. Therefore, this study aimed to examine the nutritional intake of healthcare workers working 24 h shifts on three consecutive days: the day before the shift, shift day, and the day following the shift. The aim was to evaluate the following aspects: (i) energy, macro-, and micronutrient intakes; (ii) consumption amounts from food groups; and (iii) the effects of long working hours on nutritional intake on the first day after the shift.

## 2. Materials and Methods

This study is an observational follow-up study conducted with healthcare workers working long night shifts at universities, public and private hospitals, private clinics, and 112 Emergency Aid Stations in Ankara, the capital of Turkiye. A total of 500 people, including nurses, doctors, emergency medical technicians, ambulance technicians, who worked 24 hours a day on duty, participated in this study. The shift period is 08.00 A.M. to 08.00 A.M. (24 h). The working schedule was 24 h work/24–48 h off. Throughout the article, the day worked for 24 h is defined as shift day, the day before shift day is defined as pre-shift day (off day), and the day after shift day is defined as post-shift day (off day). Healthcare workers were followed for 3 consecutive days, including shift day, pre-, and post-shift days, and food consumption records were taken on these days.

The inclusion criteria for the study were volunteer female/male health workers who were >18 years old, <60 years old, reported that they would continue their normal diet and not follow a restricted diet during the study, and could fill out the food consumption record in detail. Healthcare workers who met the following criteria were excluded from the study: women who were pregnant and breastfeeding; used nutritional supplements regularly; worked only during the day and never worked the night shift, did not want to share the information requested in collecting the data from the study, and did not want to spare time for these, were excluded from the study.

General information about the participants (age, gender, marital status, educational status, and occupation), health information (whether there is any chronic disease, smoking, and alcohol use), nutritional habits during and outside the shift (number of main/snack meals and preferred foods) information was questioned with the survey form.

Before starting the research, approval was obtained from the Gazi University Ethics Commission (Date: 6 June 2023, Research Code No: 2023-851). At the beginning of the study, the aims and method of the research were explained to all individuals included in the study, and the individuals who agreed to participate in the study read and signed a voluntary consent form.

### 2.1. Anthropometric Measurements

Body weight (kg), height (cm), waist circumference (cm), and hip circumference (cm) measurements of all individuals participating in the study were taken by the researchers. Body weight was measured using a digital scale (SECA, Hamburg, Germany) with light-as-possible clothing and no shoes. Waist circumference was measured three times at the navel level with a non-stretchable tape measure and the average of the three measurements was recorded [15,16]. Hip circumference was measured three times with a non-stretchable tape measure around the widest portion of the buttocks and the average of the three measurements was recorded [16]. Body mass index was calculated as weight/height squared (kg/m^2^) and evaluated according to the World Health Organization BMI classification [17]. In addition, individuals’ waist/hip ratio, and waist/height ratio were calculated [16,18].

### 2.2. Nutritional Assessment (3 Day Dietary Record)

Food consumption records were taken from all participants by experienced dietitians, for 3 consecutive days: the day before the shift (pre-shift, off day), the day of the shift (*shift day*), and the day after the shift (*post-shift*, off day). In the food consumption record, individuals were asked to indicate the amount and content of everything consumed (including foods, beverages, sauces, and condiments) within the 3 days. To determine the portion amounts of consumed food and beverages, the Food and Nutrition Photo Catalogue: Measurements and Quantities [19] and the Standard Food Tariffs book [20] were used. The amounts and types of foods consumed by individuals for 3 days were determined. Individuals’ daily energy, macro, and micronutrient values, and food groups were analyzed using the Nutrition Information Systems (BeBİS) program (version 9.0). Total energy, fiber, and nutrient intakes were evaluated with Dietary Reference Intakes (DRI) [21]. 

### 2.3. Physical Activity

A three-question short form, also used by Marshall et al. [22], was used to evaluate physical activity levels.

### 2.4. General Work Stress

The General Work Stress Scale, used to measure work stress, was developed by De Bruin [23] to measure nurses’ job stress. Turkish validity and reliability studies of the scale were conducted by Teleş [24]. A 5-point Likert rating was used to answer the items on the scale, which consisted of nine questions about job stress. Response options were defined as “always, frequently, sometimes, rarely, never”. Scoring was made as “1 = Never; 2 = Rarely; 3 = Sometimes; 4 = Frequently; 5 = Always”. The scale addresses the emotional, cognitive, motivational, and social consequences of the interaction between an individual and the perceived demands of the workplace. The scale score reveals the stress levels experienced or felt by the individual according to their assessment. The total score is a summary expression of the work stress experienced by the individual. High scores indicate high job stress, while low scores indicate low job stress [23].

### 2.5. Statistical Analysis

The analysis was performed using SPSS statistical software (version 28.0.0.0). Visual and analytical methods examined the normal distribution suitability of variables. Descriptive analyses were expressed as means and standard deviations (SD). The significance of differences in dietary composition between pre-shift, shift, and post-shift days was assessed using the Friedman test for repeated measures. If there was a significant difference in the Friedman test, the Wilcoxon signed-rank test was used for pairwise comparisons of the parameters. The level of statistical significance was *p* < 0.05.

## 3. Results

A total of 500 volunteer health workers, including 139 men and 361 women working night shifts in different health-related fields (shown in Appendix A), participated in this study. A total of 81.4% of the healthcare professionals had a university degree or master’s degree. More than half of the participants (60.4%) were nurses, and 72.6% of individuals stated that they had been working shifts for between 0–5 years. More than half of the individuals (56.8%) stated that they worked shifts between 6–10 times a month, 38.4% worked shifts between 1–5 times a month, and 4.8% stated that they worked >10 times a month (Table 1). More detailed information are shown in the Appendix A.

The mean and standard deviation values of individuals’ daily energy, macronutrients, fiber, and caffeine intake levels on the shift, pre-shift, and post-shift days, according to gender, are given in Table 2. The daily amount of energy received by healthcare workers on shift day, pre-shift day, and post-shift day is 2399.0 ± 739.1, 2130.2 ± 659.8, and 1749.5 ± 667.0 kcal, respectively (*p* < 0.05 for all pairwise comparisons). Like the individuals’ average daily energy intake, average daily intakes of CHO, protein, fat, saturated fat (g), and caffeine (mg) are listed as shift day > pre-shift > post-shift from highest to lowest (*p* < 0.05 for all pairwise comparisons). While cholesterol and fiber intake were similar on the shift day and pre-shift day, it was found to be statistically significantly lower on the post-shift day (*p* < 0.05) (Table 2).

Daily micronutrient intake levels of individuals on shift day, pre-shift day, and post-shift day are shown in Table 3. The average daily intake of vitamins B_1_, B_2_, niacin, B_6_, folate, and B_12_ are listed from highest to lowest as shift day > pre-shift > post-shift (*p* < 0.05 for all pairwise comparisons). While vitamin C and calcium intake amounts were similar on the shift day and pre-shift day, they were found to be statistically significantly lower on the post-shift day (*p* < 0.05). Average daily intakes of potassium, magnesium, phosphorus, iron, and zinc are listed from highest to lowest as shift day > pre-shift > post-shift (*p* < 0.05 for all pairwise comparisons) (Table 3).

Table 4 compares individuals’ daily energy and nutrient intakes with the DRI. Considering all individuals regardless of gender, fiber, folate, and calcium intake percentages were similar on pre-shift and shift days, but these values were found to be statistically significantly lower on the post-shift day (*p* < 0.05). The nutrients with the lowest DRI coverage percentages on the post-shift day were calcium (55%), fiber (65%), and folate (67%), respectively. While there was no significant difference in the percentage of women meeting their fiber and calcium needs on pre-shift and shift days, it was found that the percentage of those meeting their needs on the post-shift day was significantly lower (*p* < 0.05). The percentage of meeting folate and iron requirements in women was found to be significantly lower on the pre-shift and post-shift days compared to the shift day (*p* < 0.05 for all pairwise comparisons). While there was no significant difference in the percentage of men meeting their fiber, folate, calcium, and magnesium requirements before the shift and on the day of the shift, it was found that the percentage of those meeting the requirements after the shift was significantly lower (*p* < 0.05) (Table 4) (Figure 1 and Figure 2).

Average daily consumption amounts of individuals according to food groups are given in Table 5. Differences by gender and more detailed information are shown in the Appendix A. It was determined that the highest consumption amount from the meat group was on the shift day, followed by the pre-shift day, and the lowest was on the post-shift day (*p* < 0.05, for all pairwise comparisons). The average consumption of milk and dairy products on the shift day, pre-shift day, and post-shift day of healthcare workers was 227.7 ± 143.3 g, 205.3 ± 130.7 g and 178.8 ± 135.3 g, respectively (*p* < 0.05, for all pairwise comparisons). The amount of consumption from the bread and grain group was, from most to least, on the pre-shift day, shift day, and post-shift day (*p* < 0.05). Average daily vegetable consumption amounts were 384.4 ± 205.1 g, 362.3 ± 208.5 g and 291.1 ± 185.3 g on the shift-day, pre-shift day, and post-shift day, respectively (*p* < 0.05). The highest fruit consumption was on the pre-shift day, and the lowest fruit consumption was on the shift day (*p* < 0.05). The highest fat (53.6 ± 32.7 g) and added sugar (40.2 ± 41.1 g) intake was detected on the shift day (*p* < 0.05) (Table 5). 

## 4. Discussion

In the present study, significant differences were detected in the energy, micro-, and macronutrient intake levels between the pre-shift, post-shift, and shift days of healthcare professionals working long-hour night shifts. Intakes of energy, CHO, protein, fat, saturated fat and caffeine, vitamins B_1_, B_2_, niacin, B_6_, folate, and B_12_, potassium, magnesium, phosphorus, iron, and zinc were significantly reduced on the post-shift day compared to the pre-shift and shift days. While cholesterol, fiber, vitamin C, and calcium intakes were similar on the shift day and pre-shift day, they were found to be significantly lower on the post-shift day. It was determined that the lowest DRI coverage percentages were on post-shift days. The nutrients with the lowest DRI coverage percentages on the post-shift day were calcium (55%), fiber (total: 65%, female: 72%, male: 46%), and folate (67%), respectively.

Routine work schedules start at 8/9 a.m. and end at 5/6 p.m. on weekdays. Shift work is a way of working outside the standard schedule. It consists of evening, night, weekend, or rotating work schedules. Shift workers’ weekly working hours are significantly longer than those of day workers. Among shift work types, the longest weekly working hours are 24 h shifts and fixed night shifts. Among the factors that increase the average annual working hours of individuals in countries, the most important factor is the long working hours of night workers [25]. While many countries consider 8 h per day (40 h per week) as normal working hours, the legal maximum values are higher in some countries, including Turkiye. However, since overtime can also be done, legal limits on maximum working hours may be longer. The rules regarding maximum weekly working hours and overtime in Chile, Greece, Israel, Mexico and Poland are the same as in Turkiye [26]. In Europe, night-shift working hours are strictly regulated by law, whereas most non-European countries are not bound by any laws regarding night-shift working hours. The European Union recommends that normal working hours for night workers should not exceed 8 h (directive 2003/88/EC) [27]. Shift work, especially night-shift work that lasts long hours, requires further study, as it paves the way for the formation of four behavioral risk factors (unhealthy diet, insufficient physical activity, smoking, and alcohol use) associated with noncommunicable diseases [28]. In the European region, these four risk factors account for 77% of chronic non-communicable diseases and 86% of premature deaths. Mortality rates due to non-communicable diseases in Turkiye are like those in other countries in the WHO European Region [29]. Shift workers are exposed to some mental risks as well as physical health problems. The incidence of burnout is high in healthcare workers, due to an intense work tempo and shift work. Attention has been drawn to the relationship between eating habits and the risk of work stress and burnout [30]. It has been reported that there is a negative relationship between work stress and diet quality, especially in men. Since work stress affects the nutritional habits of employees, the evaluation of nutritional habits should be included in the scope of occupational health examinations [14]. Studies report that there are differences between men and women in the work-stress–nutrition relationship. Although work stress and high-fat and low-carbohydrate intake were reported for both genders [31], high-fat intake has also been reported only in men [32]. As an important issue, not clearly stating information about when the food consumption data of shift workers were collected and/or whether non-workdays were included in the period of food consumption records makes it difficult to compare food consumption findings in different studies. 

The evaluation of anthropometric measurements (in addition to determining the individual’s demographic characteristics, health history, eating habits, food consumption status) is one of the most important steps in determining the nutritional status of individuals [29]. Changes in mealtimes and durations during shift work are associated with an increased risk of obesity. It has been reported that consuming food late at night and for more than 14 h a day causes metabolism disorders in humans. In a study conducted on experimental animals, mice kept in a constant light/dim light cycle showed impaired glucose tolerance and increased body weight [11]. A recent meta-analysis reported that shift workers have higher body weight/BMI compared to day workers because they must work during times when they would normally be sleeping, and their circadian rhythms are disrupted [33]. It was shown that nurses have an increased risk of being overweight or obese not only due to negative eating habits, decreased physical activity levels, and increased stress [34], but also due to the overruling effect of shift work [35]. In a retrospective study of healthcare workers in Brazil, average body weight and BMI levels increased between baseline and final measurements (taken over approximately ten years). Surprisingly, no association was found between changes in body weight and BMI and shift status and occupation [36]. In another study conducted on hospital employees (health professionals and others), increased shift work was associated with increased BMI in the whole sample (OR: 3.79, 95% CI) and in health professionals (OR: 11.56, 95% CI) [37]. According to the latest Turkish Nutrition and Health Research 2019 report, conducted on approximately 13 000 people, the mean BMI for men in the 19–64 age group is 27.3 ± 5.21 kg/m^2^; waist circumference 95.0 ± 12.93 cm; hip circumference 103.6 ± 8.70 cm; waist/hip 0.91 ± 0.07; waist/height was determined as 0.55 ± 0.08. The mean BMI for women in the 19–64 age group is 28.8 ± 6.92 kg/m^2^; waist circumference 90.2 ± 15.50 cm; hip circumference 106.6 ± 12.43 cm; waist/hip 0.84 ± 0.08; waist/height is 0.57 ± 0.11 [29]. In the present study, all anthropometric measurements were found to be lower in men and women compared to the general Turkish population (with the exception that the hip circumference in men is similar). Although the mean BMI of women is higher than men in general Turkish population data [29], the opposite finding was obtained in the present 24 h shift working study sample. It can be said that the differences between the sample groups in factors, such as socio-cultural/economic and education levels, may have affected this result. However, in terms of BMI, waist/height, and waist/hip ratios, the results suggest that men working in a 24 h shift system may be at greater risk than women.

It is not surprising that shift work causes changes in individuals’ eating patterns. While working in shifts, individuals may skip more meals and consume more food outside of routine mealtimes [38]. In one study, participants reported not feeling hungry when they ate at normal mealtimes during a night shift [39]. Shift workers were unwilling to reduce their consumption of their favorite high-energy foods to reduce their energy intake. Individuals offered reasons for consuming food during the night shift, such as habit, easy access to snacks from food vending machines, and reducing fatigue [40]. A study conducted with nurses found that consuming high-glycemic-index meals during the night shift increased glycemic control and variability metrics. During the night shift, glucose profiles detected when consuming a single meal with a low glycemic index and not eating at all (fasting) were found to be similar. Additionally, when the glycemic index of meals was controlled, meal frequency during the night shift did not affect any metrics [41].

Night shift workers have higher fat intake. In the study conducted by Navruz-Varlı and Bilici [42], it was determined that nurses working in night shifts had higher energy, carbohydrate, protein, fat, and iron intakes compared to nurses working during the day. Bakirhan et al. [43] reported that emergency healthcare workers had higher energy, carbohydrate, protein, fat, cholesterol, sodium, iron, and zinc intakes on a shift day compared to rest day intakes. Additionally, they found that protein, fat, sodium, phosphorus, and zinc intakes were higher on shift days than reference values [43]. In a prospective cohort study in women, at a fat intake not exceeding reference values (≤35% Energy), even a single-day increase in the number of night shifts was associated with an increase in total fat and saturated fat intakes [14]. In a study conducted on nurses working in the accident and emergency department of a general hospital in Malta, it was reported that nurses working in shifts consumed significantly more energy (night = 1.963 kcal, rotating shift = 2.065 kcal, day = 1.722 kcal), protein, fat, and fiber, compared to those working during the day [33]. Like the commonly reported findings in studies conducted with night-shift workers, it was determined that individuals’ saturated fat intake was above recommended levels in the present study. This situation is probably explained by the following mechanism: one of the most important nutrients that provide flavor in food is fat. As a result of the circadian rhythm disorder brought about by the night shift, fats play a role as potential zeitgebers in the relationship between central and peripheral clocks. This affects eating behavior by acting directly on orexigenic centers and hedonic hunger/reward-related regions [13,44]. The shift work system affects various aspects of eating behaviors. The amount of energy, macro-, and micronutrients taken from the diet and the quality of the food consumed vary during the day when working in shifts. Changes in eating patterns, such as meal frequencies/times and meal types/contents, have been associated with sleep deprivation brought on by shift work [38]. The increased feeling of hunger due to decreased sleep time on a shift-working day causes an increase in snack consumption, resulting in higher energy, carbohydrate, and fat consumption. In the present study, the average sleep time on a shift-working day was found to be significantly lower than sleep times outside of the shift. 

While the nutritional profile found in the present study is like those previously reported in the literature for the day of night-shift work, it points to significant deficiencies in nutrient intake for the day immediately following the long-hour night shift. Individuals’ average consumption amounts of all food groups on the day after the shift were lower than before and on the day of the shift. The only exception to this was that the average amount of fruit consumed per day after a shift is higher than that consumed on the day of the shift. Although the amount of fruit consumed at home on the day after the shift increased significantly compared to the day when working the shift, it was found to be significantly lower than on the day before the shift (see Table 5). After a long, tiring, and stressful 24 h working day, in which serious decisions regarding human health are taken, healthcare workers sleep longer and have time to rest when they go home. Although individuals can access meals relatively quickly and easily from workplace cafeterias, canteens, and food vending machines on the day they work in shifts, accessing them in the home environment on the day after the shift may be more difficult due to the fatigue and burnout they experience. In addition, since healthcare workers had difficulty finding the time, strength and motivation required to procure, prepare, and cook food on the day after a shift, their energy and other nutrient intakes may have been significantly lower. In future studies on the subject, the factors affecting nutritional intake, especially on the first day after long hours of shift work, can be examined in more detail and may contribute to the planning of precaution actions. Thus, the deficiencies in calcium, magnesium (especially in men), iron (only in women), fiber (especially in men) and folate intakes, which are already common in society, can be prevented from worsening due to shift work (see Figure 1 and Figure 2).

### Strengths and Limitations

According to a recent meta-analysis, only 14% of studies on shift work and nutrition were reported to be of high quality [45]. It was emphasized that the biggest factor in this result was that the usual dietary intake within and outside the shift was not determined due to the use of a single dietary intake record (a single 24 h recall/dietary record or an FFQ). Another important point is that the studies did not clearly state the time of collection of food consumption data according to shift types and whether non-workdays were included in the collection period of the food consumption record [45]. One of the strengths of this study is that food consumption records were recorded for three consecutive days, including the day before the shift, the day of the shift and the day after the shift. In a systematic review reporting that shift work may affect the diet quality of individuals, the need to examine the duration of exposure to shift work, the duration of the working day and sleep patterns was emphasized [38]. Since the number of studies in the literature examining only long-hour night-shift workers is limited, one of the strengths of the study is that only long-hour night shifts were included in the study. Overall, the strengths of this study are its broad scope, which combines energy, macro- and micronutrient intake, and consumption amounts from the food groups of night-shift workers who work long hours (24 h per shift), an important shift work group for healthcare professionals in Turkiye. The main limitation of the study is that biochemical parameters, which are objective indicators of nutritional intake, were not examined.

## 5. Conclusions

The physical and mental pressures healthcare workers face due to their working conditions, combined with the circadian rhythm disorder that occurs due to night-shift work, pose a potential health hazard. Practices to ensure a healthier lifestyle for healthcare professionals working in night shifts are still inadequate. At this point, nutritional interventions to overcome the negative effects that night-shift work may cause are of critical importance. If environmental conditions are suitable, healthcare workers can bring healthy snacks and meals they prepare at home to work. In addition, the hospital management should ensure that healthy food varieties are available at food sales points in the hospital. To avoid the negative health consequences associated with circadian disruption, all healthcare professionals should make small, frequent meals, eaten at appropriate times, part of their daily routine. Nutrition management and health education programs should be implemented, especially for healthcare professionals working on long-hour night shifts. To ensure this, courses such as “Shift work and nutrition” can be included in the curricula of all departments studying health-related fields such as medicine and nursing. The risks of occupational health and occupational diseases can be reduced by the periodic monitoring the nutritional status of healthcare workers by dietitians. Thus, the health of healthcare workers can be protected and improved, and their performance can be increased.

Significant differences were detected the energy, micro- and macronutrient intake levels between pre-shift days, shift days, and post-shift days of healthcare professionals in the present study. The awareness of healthcare professionals should be increased regarding the decreased nutrient intake seen especially after a 24 h shift, and appropriate precautions should be taken to increase calcium, fiber, and folate intake levels.

## Figures and Tables

**Figure 1 nutrients-16-02088-f001:**
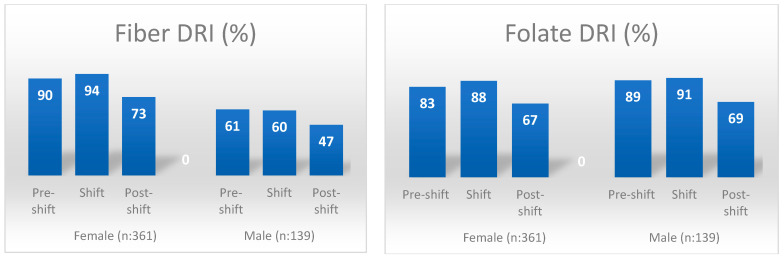
DRI percentages for fiber and folate by gender and shift days. DRI: dietary reference intakes.

**Figure 2 nutrients-16-02088-f002:**
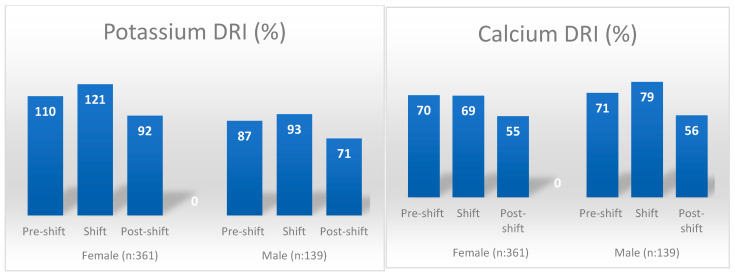
Minerals with the lowest DRI percentage by gender and shift days. DRI: dietary reference intakes.

**Table 1 nutrients-16-02088-t001:** Socio-demographic characteristics of individuals.

Variables	Total (N:500)	Female (n:361)	Male (n:139)	
	Number (%) or Mean ± SD	Number (%) or Mean ± SD	Number (%) or Mean ± SD	
Age (years)	30.7 ± 7.01	30.5 ± 7.0	31.2 ± 7.1	t = 0.901 *p* = 0.368
Anthropometric measurements				
Body weight (kg)	68.6 ± 12.8	64.4 ± 10.6	79.6 ± 11.4	t = 14.129 <0.001 **
Height (cm)	167.0 ± 11.6	164.1 ± 8.1	175.6 ± 7.5	t = 14.479 <0.001 **
BMI (kg/m^2^)	24.4 ± 3.6	23.8 ± 3.6	25.8 ± 3.3	t = 5.677 <0.001 **
Waist circumference (cm)	82.9 ± 13.4	79.8 ± 12.6	91.1 ± 11.9	t = 9.144 <0.001 **
Hip circumference (cm)	98.2 ± 13.0	96.2 ± 12.0	103.7 ± 12.0	t = 6.258 <0.001 **
Waist/height	0.49 ± 0.08	0.48 ± 0.08	0.52 ± 0.06	t = 3.935 <0.001 **
Waist/hip	0.85 ± 0.11	0.83 ± 0.11	0.88 ± 0.09	t = 4.506 <0.001 **
Consumption of alcoholic beverages				
Consumes	54 (10.8)	35 (9.7)	19 (13.7)	*X*^2^ = 2.258 *p* = 0.133
Does not consume	446 (89.2)	326 (90.3)	120 (86.3)	
Mean frequency of alcoholic beverage consumption (times/month)	1.9 ± 1.4	1.8 ± 1.2	2.0 ± 1.8	t = 0.419 *p* = 0.677
Alcoholic beverage consumption amount (mL)	306.4 ± 190.4	353.5 ± 171.0	226.3 ± 199.0	t = −2.336 *p* = 0.023
Marital status				
Married	244 (48.8)	177 (49.0)	67 (48.2)	*X*^2^ = 2.611 *p* = 0.271
Single	256 (51.2)	184 (51.0)	72 (51.8)	
Educational status				
High school	27 (5.4)	18 (5.0)	9 (6.5)	*X*^2^ = 6.319 *p* = 0.097
Associate degree	116 (23.2)	74 (20.5)	42 (30.2)	
License	293 (68.6)	220 (60.9)	73 (52.5)	
Postgraduate	64 (12.8)	49 (13.6)	15 (10.8)	
Occupation				
Nurse	302 (60.4)	229 (63.4)	73 (52.5)	*X*^2^ = 6.315 *p* = 0.097
Doctor	48 (9.6)	29 (8.1)	19 (13.7)	
Emergency medical technician	103 (20.6)	70 (19.4)	33 (23.7)	
Ambulance technician	47 (9.4)	33 (9.1)	14 (10.1)	
Total time worked in shifts (years)				
0–5	363 (72.6)	251 (69.5)	112 (80.6)	*X*^2^ = 3.308 *p* = 0.191
6–10	71 (14.2)	53 (14.7)	18 (12.9)	
>10	66 (13.2)	57 (15.8)	9 (6.5)	
Number of shifts (number/month)				
1–5	192 (38.4)	136 (37.7)	56 (40.3)	*X*^2^ = 0.199 *p* = 0.905
6–10	284 (56.8)	208 (57.6)	76 (54.7)	
>10	24 (4.8)	17 (4.7)	7 (5.0)	
Sleep time (hour/day)				
On-shift	3.8 ± 1.5	3.9 ± 1.4	3.7 ± 1.6	t = −0.906 *p* = 0.366
Non-shift day	7.6 ± 1.5	7.6 ± 1.6	7.5 ± 1.5	
Does changing sleep patterns during a shift affect nutrition?				
Yes	410 (82.0)	302 (83.7)	108 (77.7)	*X*^2^ = 2.414 *p* = 0.120
No	90 (18.0)	59 (16.3)	31 (22.3)	
How does changing sleep patterns during a shift affect nutrition?				
Increases	261 (52.2)	202 (56.0)	59 (42.4)	*X*^2^ = 6.611 *p* = 0.037 *
Reduces	148 (29.6)	98 (27.1)	50 (36.0)	
Does not change	91 (18.2)	61 (16.9)	30 (21.6)	
Number of main meals during shift	2.5 ± 0.7	2.5 ± 0.7	2.5 ± 0.7	t = 0.212 *p* = 0.832
Number of snacks during shift	2.1 ± 1.1	2.2 ± 1.1	2.1 ± 1.1	t = −1.111 *p* = 0.267
Physical activity status				
“Inadequately” active	463 (92.6)	342 (94.7)	121 (87.1)	*X*^2^ = 2.122 *p* = 0.145
“Adequately” active	37 (7.4)	19 (5.3)	18 (12.9)	

* *p* < 0.05, ** *p* < 0.001, SD: standard deviation, BMI: body mass index.

**Table 2 nutrients-16-02088-t002:** Individuals’ daily energy, macronutrients, fiber, and caffeine intakes on shift, pre-, and post-shift days.

Variables	Total (N:500)			Female (n:361)			Male (n:139)		
	Pre-Shift	Shift	Post-Shift	Pre-Shift	Shift	Post-Shift	Pre-Shift	Shift	Post-Shift
	Mean ± SD	Mean ± SD	Mean ± SD	Mean ± SD	Mean ± SD	Mean ± SD	Mean ± SD	Mean ± SD	Mean ± SD
Energy (kcal)	2130.2 ± 659.8 ^a^	2399.0 ± 739.1 ^b^	1749.5 ± 667.0 ^c^	2096.6 ± 645.0 ^a^	2389.3 ± 738.4 ^b^	1757.6 ± 658.1 ^c^	2217.5 ± 691.6 ^a^	2424.5 ± 743.1 ^b^	1728.4 ± 691.6 ^c^
CHO (g)	230.6 ± 81.6 ^a^	256.5 ± 93.5 ^b^	192.4 ± 78.4 ^c^	225.9 ± 78.0 ^a^	256.2 ± 94.1 ^b^	194.1 ± 77.4 ^c^	242.8 ± 89.4 ^a^	257.4 ± 92.2 ^a^	188.1 ± 81.2 ^b^
CHO (%)	44.4 ± 8.2 ^a,b^	43.7 ± 7.4 ^b^	45.4 ± 8.7 ^a^	44.3 ± 8.3 ^a^	43.9 ± 7.5 ^a^	45.6 ± 8.6 ^b^	44.6 ± 7.9 ^a^. ^b^	43.2 ± 6.9 ^a^	45.0 ± 9.0 ^b^
Protein (g)	75.8 ± 26.3 ^a^	85.2 ± 26.9 ^b^	59.8 ± 25.4 ^c^	75.6 ± 27.4 ^a^	84.5 ± 27.1 ^b^	59.5 ± 25.5 ^c^	76.2 ± 23.2 ^a^	87.0 ± 26.3 ^b^	60.6 ± 25.1 ^c^
Protein (%)	14.9 ± 3.7	14.9 ± 3.7	14.5 ± 4.2	15.0 ± 3.6 ^a^	14.8 ± 3.6 ^a,b^	14.3 ± 4.0 ^b^	14.6 ± 3.9	15.1 ± 3.9	15.1 ± 4.6
Fat (g)	97.8 ± 39.1 ^a^	112.1 ± 43.1 ^b^	79.9 ± 38.8 ^c^	99.1 ± 38.8 ^a^	111.4 ± 43.1 ^b^	80.2 ± 38.6 ^c^	102.0 ± 39.7 ^a^	113.7 ± 43.4 ^b^	79.1 ± 39.6 ^c^
Fat (%)	40.5 ± 7.8 ^a,b^	41.2 ± 7.3 ^b^	39.9 ± 8.6 ^a^	40.4 ± 7.7	41.1 ± 7.4	40.0 ± 8.3	40.7 ± 8.2	41.5 ± 7.2	39.8 ± 9.4
SFA (g)	31.5 ± 13.9 ^a^	35.9 ± 14.9 ^b^	25.5 ± 13.3 ^c^	31.4 ± 14.3 ^a^	35.3 ± 14.7 ^b^	25.7 ± 13.5 ^c^	31.8 ± 12.9 ^a^	37.3 ± 15.2 ^b^	24.9 ± 12.5 ^c^
MUFA (g)	32.2 ± 12.8 ^a^	36.7 ± 14.0 ^b^	26.0 ± 12.8 ^c^	31.7 ± 12.7 ^a^	36.4 ± 14.0 ^b^	26.1 ± 13.0 ^c^	33.6 ± 13.1 ^a^	37.5 ± 14.0 ^b^	25.7 ± 12.4 ^c^
PUFA (g)	25.1 ± 14.1 ^a^	29.7 ± 15.7 ^b^	21.2 ± 13.9 ^c^	24.4 ± 13.7 ^a^	29.9 ± 15.7 ^b^	21.3 ± 13.5 ^c^	27.1 ± 14.9 ^a^	29.2 ± 15.6 ^a^	21.2 ± 15.0 ^b^
Cholesterol (mg)	359.8 ± 197.5 ^a^	358.4 ± 187.0 ^a^	268.9 ± 167.9 ^b^	362.5 ± 202.3 ^a^	354.2 ± 190.9 ^a^	264.6 ± 170.5 ^b^	352.8 ± 184.9 ^a^	369.3 ± 176.7 ^a^	279.9 ± 161.1 ^b^
Fiber (g)	22.5 ± 9.1 ^a^	23.2 ± 8.5 ^a^	17.9 ± 8.7 ^b^	22.4 ± 9.2 ^a^	23.4 ± 8.6 ^a^	18.1 ± 8.7 ^b^	23.0 ± 8.8 ^a^	22.6 ± 8.2 ^a^	17.5 ± 8.5 ^b^
Caffeine (mg)	67.7 ± 68.8 ^a^	81.8 ± 77.6 ^b^	51.6 ± 58.5 ^c^	65.9 ± 69.6 ^a^	83.6 ± 79.9 ^b^	52.8 ± 62.1 ^c^	72.5 ± 66.6 ^a^	77.2 ± 71.1 ^a^	48.4 ± 48.2 ^b^

^a,b,c^ Different letters indicate statistical significance between shift, pre-, and post-shift days, *p* < 0.05. SD: Standard deviation, CHO: Carbohydrate, SFA: saturated fatty acid, MUFA: Monounsaturated fatty acid, PUFA: Polyunsaturated fatty acid.

**Table 3 nutrients-16-02088-t003:** Daily micronutrient intake levels of individuals on shift day, pre-shift day, and post-shift day.

Variables	Total (N:500)			Female (n:361)			Male (n:139)		
	Pre-Shift	Shift	Post-Shift	Pre-Shift	Shift	Post-Shift	Pre-Shift	Shift	Post-Shift
	Mean ± SD	Mean ± SD	Mean ± SD	Mean ± SD	Mean ± SD	Mean ± SD	Mean ± SD	Mean ± SD	Mean ± SD
Vitamin A (mcg)	1395.0 ± 1963.4 ^a^	1673.5 ± 2730.1 ^a^	1059.1 ± 800.4 ^b^	1346.3 ± 1925.8 ^a^	1700.9 ± 2960.7 ^a^	1055.8 ± 799.1 ^b^	1521.2 ± 2059.6 ^a^	1602.2 ± 2019.0 ^a^	1067.8 ± 806.5 ^b^
Vitamin E (mg)	25.5 ± 13.3 ^a^	28.0 ± 13.7 ^b^	20.6 ± 13.5 ^c^	24.4 ± 12.3 ^a^	28.2 ± 14.3 ^b^	20.3 ± 13.2 ^c^	28.5 ± 15.1 ^a^	27.4 ± 12.3 ^a^	21.3 ± 14.2 ^b^
Vitamin K (mcg)	112.2 ± 153.9 ^a^	136.0 ± 195.2 ^b^	110.9 ± 165.6 ^a^	106.3 ± 146.0	125.4 ± 168.7	113.3 ± 169.9	127.5 ± 172.3 ^a,b^	163.6 ± 250.0 ^a^	104.6 ± 154.4 ^b^
Thiamin (mg)	1.0 ± 0.3 ^a^	1.1 ± 0.4 ^b^	0.8 ± 0.3 ^c^	1.0 ± 0.3 ^a^	1.1 ± 0.4	0.8 ± 0.3 ^c^	1.0 ± 0.4 ^a^	1.1 ± 0.4 ^a^	0.8 ± 0.4 ^b^
Riboflavin (mg)	1.5 ± 0.6 ^a^	1.6 ± 0.7 ^b^	1.2 ± 0.5 ^c^	1.5 ± 0.6 ^a^	1.5 ± 0.7 ^a^	1.2 ± 0.5 ^b^	1.5 ± 0.6 ^a^	1.6 ± 0.6 ^a^	1.2 ± 0.4 ^b^
Niacin (mg)	16.2 ± 8.3 ^a^	19.1 ± 9.7 ^b^	12.6 ± 8.1 ^c^	16.0 ± 8.4 ^a^	19.2 ± 10.0 ^b^	12.4 ± 7.8 ^c^	16.5 ± 8.3 ^a^	18.9 ± 8.9 ^b^	12.9 ± 8.7 ^c^
Vitamin B_6_ (mg)	1.5 ± 0.6 ^a^	1.7 ± 0.6 ^b^	1.2 ± 0.5 ^c^	1.5 ± 0.6 ^a^	1.6 ± 0.6 ^b^	1.2 ± 0.5 ^c^	1.6 ± 0.6 ^a^	1.7 ± 0.6 ^a^	1.3 ± 0.6 ^b^
Folate (mcg)	336.8 ± 144.8 ^a^	355.1 ± 157.8 ^b^	270.8 ± 128.4 ^c^	330.1 ± 140.1 ^a^	352.1 ± 161.6 ^b^	269.1 ± 129.7 ^c^	354.3 ± 155.4 ^a^	362.9 ± 147.6 ^a^	275.2 ± 125.4 ^b^
Vitamin B_12_ (mcg)	5.9 ± 7.6 ^a^	7.3 ± 9.9 ^b^	4.4 ± 4.1 ^c^	6.0 ± 7.7 ^a^	7.3 ± 10.8 ^a^	4.4 ± 4.3 ^b^	5.6 ± 7.4 ^a, b^	7.1 ± 7.1 ^a^	4.3 ± 3.4 ^b^
Vitamin C (mg)	142.0 ± 87.2 ^a^	145.9 ± 80.1 ^a^	117.9 ± 80.5 ^b^	140.5 ± 91.4 ^a^	145.9 ± 80.7 ^a^	117.5 ± 81.6 ^b^	145.8 ± 75.4 ^a^	146.1 ± 78.6 ^a^	119.0 ± 77.8 ^b^
Potassium (mg)	2883.5 ± 934.4 ^a^	3148.6 ± 1006.2 ^b^	2393.0 ± 934.3 ^c^	2854.2 ± 934.4 ^a^	3140.9 ± 1006.9 ^b^	2387.1 ± 932.7 ^c^	2959.7 ± 933.4 ^a^	3168.5 ± 1007.6 ^a^	2408.4 ± 941.9 ^b^
Sodium (mg)	1667.2 ± 948.7 ^a^	1539.5 ± 747.8 ^b^	1260.8 ± 771.8 ^c^	1650.1 ± 878.2 ^a^	1502.6 ± 751.2 ^b^	1245.6 ± 753.3 ^c^	1711.5 ± 1113.7 ^a^	1635.4 ± 732.8 ^a^	1300.3 ± 819.2 ^b^
Calcium (mg)	702.4 ± 284.4 ^a^	713.0 ± 263.4 ^a^	556.3 ± 243.0 ^b^	698.4 ± 290.4 ^a^	695.4 ± 263.4 ^a^	555.2 ± 244.5 ^b^	712.7 ± 268.9 ^a^	758.6 ± 258.7 ^a^	559.3 ± 240.0 ^b^
Magnesium (mg)	303.3 ± 106.7 ^a^	330.6 ± 109.7 ^b^	243.1 ± 101.2 ^c^	301.0 ± 103.0 ^a^	329.8 ± 109.2 ^b^	244.8 ± 100,1 ^c^	309.3 ± 116.1 ^a^	332.6 ± 111.4 ^a^	238.6 ± 104.1 ^b^
Phosphorus (mg)	1172.5 ± 368.6 ^a^	1269.1 ± 388.0 ^b^	931.8 ± 344.0 ^c^	1165.8 ± 370.4 ^a^	1260.5 ± 391.6 ^b^	934.1 ± 351.2 ^c^	1189.9 ± 364.7 ^a^	1291.4 ± 378.9 ^b^	925.8 ± 325.7 ^c^
Iron (mg)	12.0 ± 4.5 ^a^	13.4 ± 4.9 ^b^	9.6 ± 4.4 ^c^	11.9 ± 4.5 ^a^	13.5 ± 5.0 ^b^	9.6 ± 4.3 ^c^	12.3 ± 4.6 ^a^	13.3 ± 4.6 ^a^	9.6 ± 4.5 ^b^
Zinc (mg)	11.6 ± 4.8 ^a^	13.2 ± 4.3 ^b^	9.1 ± 4.6 ^c^	11.6 ± 5.0 ^a^	13.0 ± 4.2 ^b^	9.2 ± 4.8 ^c^	11.7 ± 4.1 ^a^	13.6 ± 4.4 ^b^	9.0 ± 4.0 ^c^

^a,b,c^ Different letters indicate statistical significance between shift, pre-shift, and post-shift days, *p* < 0.05,. SD: standard deviation.

**Table 4 nutrients-16-02088-t004:** Comparison of individuals’ daily energy and nutrient intakes with DRI (%).

Variables	Total (N:500)			Female (n:361)			Male (n:139)		
	Pre-Shift	Shift	Post-Shift	Pre-Shift	Shift	Post-Shift	Pre-Shift	Shift	Post-Shift
	Mean ± SD	Mean ± SD	Mean ± SD	Mean ± SD	Mean ± SD	Mean ± SD	Mean ± SD	Mean ± SD	Mean ± SD
Energy (kcal)	98.4 ±3 2.7 ^a^	111.0 ± 37.2 ^b^	81.1 ± 33.1 ^c^	104.8 ± 32.9 ^a^	119.2 ± 37.1 ^b^	87.8 ± 33.2 ^c^	81.9 ± 25.8 ^a^	89.6 ± 27.9 ^b^	63.9 ± 26.2 ^c^
CHO (g)	177.4 ± 62.7 ^a^	197.3 ± 71.9 ^b^	148.0 ± 60.3 ^c^	173.8 ± 60.0 ^a^	197.1 ± 72.3 ^b^	149.3 ± 59.5 ^c^	186.8 ± 68.7 ^a^	198.0 ± 70.9 ^a^	144.7 ± 62.4 ^b^
Protein (g)	156.6 ± 56.6 ^a^	175.8 ± 57.3 ^b^	123.6 ± 53.6 ^c^	164.4 ± 59.6 ^a^	183.7 ± 59.0 ^b^	129.5 ± 55.5 ^c^	136.1 ± 41.5 ^a^	155.4 ± 47.1 ^b^	108.3 ± 44.8 ^c^
Fiber (g)	81.9 ± 36.5 ^a^	84.7 ± 35.1 ^a^	65.5 ± 34.5 ^b^	89.9 ± 37.2 ^a^	94.1 ± 34.6 ^a^	72.7 ± 35.1 ^b^	61.1 ± 24.6 ^a^	60.2 ± 22.5 ^a^	46.8 ± 24.2 ^b^
Vitamin A (mcg)	185.8 ± 263.0 ^a^	224.9 ± 379.2 ^a^	141.8 ± 108.7 ^b^	192.3 ± 275.1 ^a^	242.9 ± 422.9 ^a^	150.8 ± 114.1 ^b^	169.0 ± 228.8 ^a^	178.0 ± 224.3 ^a^	118.6 ± 89.6 ^b^
Vitamin E (mg)	170.3 ± 88.7 ^a^	187.0 ± 91.9 ^b^	137.6 ± 90.4 ^c^	162.6 ± 82.4 ^a^	188.5 ± 95.4 ^b^	135.9 ± 88.5 ^c^	190.3 ± 100.9 ^a^	183.2 ± 82.4 ^a^	142.0 ± 95.1 ^b^
Thiamin (mg)	89.6 ± 30.8 ^a^	98.0 ± 32.4 ^b^	73.0 ± 31.1 ^c^	90.8 ± 30.6 ^a^	99.8 ± 33.3 ^b^	74.7 ± 31.5 ^c^	86.6 ± 31.1 ^a^	93.3 ± 29.8 ^a^	68.6 ± 29.7 ^b^
Riboflavin (mg)	134.7 ± 61.0 ^a^	134.7 ± 61.0 ^b^	100.4 ± 40.2 ^c^	139.4 ± 65.9 ^a^	139.4 ± 65.9 ^a^	104.7 ± 41.7 ^b^	122.4 ± 44.2 ^a^	122.4 ± 44.2 ^a^	89.1 ± 33.8 ^b^
Niacin (mg)	111.6 ± 58.1 ^a^	132.2 ± 68.2 ^b^	86.9 ± 55.7 ^c^	114.8 ± 60.1 ^a^	137.6 ± 71.8 ^b^	89.2 ± 56.1 ^c^	103.1 ± 51.8 ^a^	118.2 ± 55.7 ^b^	80.9 ± 54.4 ^c^
Vitamin B_6_ (mg)	119.4 ± 47.6 ^a^	131.1 ± 50.7 ^b^	96.4 ± 45.7 ^c^	118.3 ± 48.3 ^a^	130.3 ± 49.7 ^b^	95.3 ± 44.6 ^c^	122.2 ± 45.9 ^a^	133.1 ± 53.2 ^a^	99.2 ± 48.6 ^b^
Folate (mcg)	84.2 ± 36.2 ^a^	88.7 ±3 9.4 ^a^	67.7 ± 32.1 ^b^	82.5 ± 35.0 ^a^	88.0 ± 40.4 ^b^	67.2 ± 32.4 ^c^	88.5 ± 38.8 ^a^	90.7 ± 36.9 ^a^	68.8 ± 31.3 ^b^
Vitamin B_12_ (mcg)	248.3 ± 318.8 ^a^	305.6 ± 413.0 ^b^	184.1 ± 171.0 ^c^	253.0 ± 322.2 ^a^	308.1 ± 450.4 ^a^	184.6 ± 181.0 ^b^	236.0 ± 310.7 ^a,b^	299.3 ± 295.9 ^a^	182.8 ± 142.3 ^b^
Vitamin C (mg)	180.3 ± 113.1 ^a^	185.6 ± 103.4 ^a^	149.9 ± 103.5 ^b^	187.3 ± 121.9 ^a^	194.5 ± 107.7 ^a^	156.7 ± 108.8 ^b^	162.0 ± 83.8 ^a^	162.4 ± 87.4 ^a^	132.2 ± 86.5 ^b^
Potassium (mg)	103.4 ± 35.2 ^a^	113.1 ± 38.4 ^b^	85.9 ± 35.0 ^c^	109.7 ± 35.9 ^a^	120.8 ± 38.7 ^b^	91.8 ± 35.8 ^c^	87.0 ± 27.4 ^a^	93.1 ± 29.6 ^a^	70.8 ± 27.7 ^b^
Sodium (mg)	111.1 ± 63.2 ^a^	102.6 ± 49.8 ^b^	84.0 ± 51.4 ^c^	110.0 ± 58.5 ^a^	100.1 ± 50.0 ^b^	83.0 ± 50.2 ^c^	114.1 ± 74.2 ^a^	109.0 ± 48.8 ^a^	86.6 ± 54.6 ^b^
Calcium (mg)	70.1 ± 28.4 ^a^	71.1 ± 26.3 ^a^	55.5 ± 24.3 ^b^	69.7 ± 29.0 ^a^	69.3 ± 26.3 ^a^	55.4 ± 24.4 ^b^	71.2 ± 26.8 ^a^	78.8 ± 25.8 ^a^	55.9 ± 24.0 ^b^
Magnesium (mg)	90.3 ± 32.8 ^a^	98.6 ± 34.7 ^b^	72.5 ± 31.4 ^c^	95.9 ± 32.6 ^a^	105.2 ± 35.0 ^b^	78.0 ± 31.9 ^c^	75.7 ± 28.4 ^a^	81.5 ± 27.3 ^a^	58.4 ± 25.3 ^b^
Phosphorus (mg)	167.5 ± 52.6 ^a^	181.3 ± 55.4 ^b^	133.1 ± 49.1 ^c^	166.5 ± 52.9 ^a^	180.0 ± 55.9 ^b^	133.4 ± 50.1 ^c^	169.9 ± 52.1 ^a^	184.4 ± 54.1 ^b^	132.2 ± 46.5 ^c^
Iron (mg)	91.9 ± 55.2 ^a^	101.1 ± 56.2 ^b^	72.7 ± 47.2 ^c^	67.9 ± 28.9 ^a^	76.1 ± 28.8 ^b^	54.4 ± 25.3 ^c^	154.3 ± 58.2 ^a^	166.2 ± 57.9 ^a^	120.1 ± 56.9 ^b^
Zinc (mg)	135.0 ± 59.8 ^a^	152.7 ± 53.2 ^b^	106.1 ± 57.4 ^c^	145.8 ± 63.2 ^a^	163.6 ± 53.6 ^b^	115.5 ± 61.2 ^c^	106.8 ± 37.7 ^a^	124.3 ± 40.0 ^b^	81.8 ± 36.3 ^c^

^a,b,c^ Different letters indicate statistical significance between shift, pre-, and post-shift, *p* < 0.05. SD: Standard deviation, CHO: Carbohydrate.

**Table 5 nutrients-16-02088-t005:** Mean daily consumption amounts of food groups.

Food Groups	Total (N:500)		
	Pre-Shift	Shift	Post-Shift
	Mean ± SD	Mean ± SD	Mean ± SD
Meat and meat products, eggs, legumes, and nuts/seeds (g)	181.9 ± 115.3 ^a^	218.3 ± 107.7 ^b^	145.8 ± 111.5 ^c^
Milk and dairy products (g)	205.3 ± 130.7 ^a^	227.7 ± 143.3 ^b^	178.8 ± 135.3 ^c^
Bread and cereals (g)	175.6 ± 96.9 ^a^	169.4 ± 91.5 ^b^	131.6 ± 81.3 ^c^
Vegetables (g)	362.3 ± 208.5 ^a^	384.4 ± 205.1 ^b^	291.1 ± 185.3 ^c^
Fruits (g)	121.5 ± 116.2 ^a^	98.8 ± 98.3 ^b^	105.4 ± 120.6 ^c^
Fats (g)	49.5 ± 32.3 ^a^	53.6 ± 32.7 ^b^	42.1 ± 32.1 ^c^
Added sugars (g)	32.6 ± 34.5 ^a^	40.2 ± 41.1 ^b^	29.5 ± 33.1 ^c^

^a,b,c^ Different letters indicate statistical significance between shift, pre-shift, and post-shift, *p* < 0.05. SD: Standard deviation.

## Data Availability

Data can be shared upon request from the corresponding author.

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
