# Peer review of "The Effect of 24 h Shift Work on the Nutritional Status of Healthcare Workers: An Observational Follow-Up Study from Türkiye"

_nutrients, 2024, doi:10.3390/nu16132088_

Round 1
Reviewer 1 Report
Comments and Suggestions for Authors
In the study, the authors collected and analyzed dietary data from 500 healthcare workers for three consecutive days: pre-shift, shift day, and post-shift. Significant differences were found in the energy, micronutrient, and macronutrient intake levels between pre-shift, shift day, and post-shift days. The results obtained are new and have important implications for monitoring the nutritional status and health of shift workers.
The manuscript is well written, with a detailed description of the background, a clear indication of the purpose and design of the study, and a presentation of the findings. Therefore, I believe that the manuscript can be accepted after minor improvements.
Abstract:
1. DRI should be written in full.
Results:
2.Anthropometric characteristics are recommended to be transferred to the main text from the supplementary file.
3.It is not clear why the results presented do not include data on the consumption of alcoholic beverages and sweet sodas, which may affect circadian rhythms.
Discussion:
4. Please analyze and compare the anthropometric characteristics of the survey participants with those of the general Turkish or European population.
5. Please interpret and suggest possible reasons for the increased food intake on shift days.
6. Please interpret and suggest possible reasons for the decreased food intake on post-shift days.
Author Response
|
1. Summary |
|
|
|
Thank you very much for taking the time to review this manuscript. Please find the detailed responses below and the corresponding revisions highlighted in the re-submitted files. |
||
|
2. Point-by-point response to Comments and Suggestions |
||
|
Comments 1: DRI should be written in full. |
||
|
Response 1: Thank you for pointing this out. We agree with this comment. Therefore, we have written DRI in full in the abstract section, in line 21. |
||
|
Comments 2: Anthropometric characteristics are recommended to be transferred to the main text from the supplementary file. |
||
|
Response 2: We moved the anthropometric measurement data from the supplementary section to Table 1 in the main article. |
||
|
Comments 3: It is not clear why the results presented do not include data on the consumption of alcoholic beverages and sweet sodas, which may affect circadian rhythms. |
||
|
Response 3: Thank you for highlighting a point we missed with your precious suggestion. Although data on alcoholic beverage consumption are available, we think it is a deficiency that is not shown in the article. The relevant data are added to Table 1 in the main article. Carbonated beverage consumption data were queried for shift days and non-shift days, the relevant data are shown in supplementary material Table S4. Also, added sugar data, which also includes sugar from carbonated drinks, is shown in Table 5 in the main article. |
||
|
Comments 4: Please analyze and compare the anthropometric characteristics of the survey participants with those of the general Turkish or European population. |
||
|
Response 4: Thank you for your valuable suggestion. We compare the anthropometric characteristics of the survey participants with those of the general Turkish population, in line 306-320. |
||
|
Comments 5: Please interpret and suggest possible reasons for the increased food intake on shift days. |
||
|
Response 5: Thank you for your suggestion, which we think will contribute significantly to the development of the article. Possible reasons for increased food intake on shift days are commented, in line 354-361. |
||
|
Comments 6: Please interpret and suggest possible reasons for the decreased food intake on post-shift days. |
||
|
Response 6: Thank you for your suggestion, which we think will contribute significantly to the development of the article. Possible reasons for reduced food intake in post-shift days are commented, in line 365-385. |
||

Reviewer 2 Report
Comments and Suggestions for Authors
The aim of the study was to determine the diet, and especially the daily intake of macro- and micronutrients, by health care workers working 24-hour shifts for three consecutive days: the day before the shift, the day of the shift and the day after the shift. An assessment was made of (i) energy, macro- and micronutrient intake, (ii) consumption of individual food groups and (iii) the impact of long working hours on nutrient intake on the first day after the shift. The study was carried out very carefully using appropriate tools. The authors showed that on the day off after the 24-hour shift, there was a decrease in the consumption of individual food groups, and with it a decrease in the intake of energy, CHO, protein, fat, saturated fat and caffeine, vitamins B1, B2, niacin, B6, folate, and B12, potassium, magnesium, phosphorus, iron, zinc, cholesterol, fiber, vitamin C, and calcium intakes. As a result, on the post-shift day off there were the lowest DRI coverage percentages of these products.
In the discussion, the authors only cite other studies on the diet of shift workers. It lacks even an attempt to consider what could have been the cause and what was the probable mechanism of such a serious change in the diet observed in our own research on a day off after the change.
Without such a supplement to the discussion, the article has the characteristics of a research report and not a scientific article worthy of publication.
Author Response
|
1. Summary |
|
|
|
Thank you very much for taking the time to review this manuscript. Please find the detailed responses below and the corresponding revisions highlighted in the re-submitted files. |
||
|
2. Point-by-point response to Comments and Suggestions |
||
|
Comments 1: In the discussion, the authors only cite other studies on the diet of shift workers. It lacks even an attempt to consider what could have been the cause and what was the probable mechanism of such a serious change in the diet observed in our own research on a day off after the change. Without such a supplement to the discussion, the article has the characteristics of a research report and not a scientific article worthy of publication. |
||
|
Response 1: Thank you for your suggestion, which we think will contribute significantly to the development of the article. Possible reasons for increased food intake on shift days are commented, in line 354-361. Possible reasons for reduced food intake in post-shift days are commented, in line 365-385. |
||